# Modification-Considering Value Learning for Reward Hacking Mitigation in RL

## Abstract

Reinforcement learning agents can exploit poorly designed reward signals to achieve high apparent returns while failing to satisfy the intended objective, a failure mode known as reward hacking. We address this in standard value-based RL with Modification-Considering Value Learning (MCVL), a safeguard that treats each learning update as a decision to evaluate. When a new transition arrives, the agent forecasts two futures: one that learns from the transition and one that does not. It then scores both using its current learned return estimator, which combines predicted rewards with a value-function bootstrap, and accepts the transition only if admission does not decrease that score. We provide DDQN- and TD3-based implementations and show that MCVL prevents reward hacking across diverse environments, including AI Safety Gridworlds and a modified MuJoCo Reacher task, while continuing to improve the intended objective. To our knowledge, MCVL is the first practical implementation of an agent that evaluates its own modifications, offering a step toward robust defenses against reward hacking.

## 1 Introduction

Optimizing poorly defined or incomplete rewards can push RL agents toward unintended behaviors, leading to *reward hacking* (Skalse et al., 2022). For instance, an agent tasked with stacking blocks may learn to flip blocks if the reward is based on the height of the bottom face (Popov et al., 2017). As RL systems scale to safety-critical applications (e.g., autonomous driving (Kiran et al., 2021) or medical diagnostics (Ghesu et al., 2017)), ensuring reliable and safe behavior becomes increasingly important. Reward hacking can become more prevalent as models grow in complexity (Pan et al., 2022), which also affects large language models where RL is used for post-training (Denison et al., 2024; OpenAI, 2024). A common mitigation constrains policy updates around a trusted reference (Laidlaw et al., 2024), often at a cost to optimality.

A complementary safeguard is to *optimize what the agent currently values* while being conservative about changing those values, an idea discussed as *current utility optimization* (Orseau & Ring, 2011; Hibbard, 2012; Everitt et al., 2016; 2021). These works largely lack a practical algorithm implementing this idea. We fill this research gap by investigating whether individual transitions can be predictive of reward hacking in the context of value-based RL. Our method, *Modification-Considering Value Learning (MCVL)*, wraps a standard off-policy learner and treats each update as a candidate modification. For a newly observed transition, the agent forecasts two scenarios: one in which it learns from the transition and one that ignores it. Then MCVL evaluates the resulting policies using its *current* learned return estimator, an $n$-step bootstrapped return combining a learned reward model with a value-function bootstrap. The transition is accepted only if its inclusion does not reduce this score relative to continuing training without it. Intuitively, MCVL blocks updates that, according to the agent's current return estimator, would shift behavior toward undesirable strategies (e.g., reward tampering rather than task completion), thereby avoiding reliance on an external oracle or a predefined safe policy.

We instantiate MCVL with DDQN and TD3. We present MCVL as a practical instantiation of the idea that an agent should evaluate its own modifications. To keep the evaluation controlled, we focus on simpler environments and defer larger-scale applications, such as language modeling, to future work. To make the predictions of the reward model and critic meaningful from the beginning, we pretrain them on a small seed buffer without reward hacking transitions. For didactic gridworlds, where

undirected exploration quickly discovers hacking, we collect this buffer in a *Safe* variant that matches observations, actions, and proxy rewards but removes the hacking affordance, i.e., makes hacking states unreachable. For continuous control, where random exploration is unlikely to encounter hacks, we pretrain directly in the *Full* environment with a small dataset collected via a random policy. Under these conditions, MCVL prevents reward hacking in AI Safety Gridworlds (Leike et al., 2017) and a modified Gymnasium Reacher environment (Towers et al., 2024) while continuing to improve the intended performance. In all cases, final training and evaluation are conducted in the unmodified Full environments so that hacking opportunities remain available during learning and testing.

Our contributions are:

- A simple *forecast-and-score* safeguard for off-policy value-based RL that admits a transition only when it does not reduce the agent's *current* bootstrapped-return estimation.

- Implementations for discrete and continuous control environments (MC-DDQN, MC-TD3).

- Empirical evidence across multiple environments, including AI Safety Gridworlds and MuJoCo Reacher, that MCVL prevents reward hacking while reaching Oracle performance.

## 2 PRELIMINARIES

We consider a Markov decision process (MDP) $(S, A, P, R, \rho, \gamma)$ with state space $S$, action space $A$, transition model $P(s'|s, a) \in [0, 1]$, reward function $R : S \times A \to \mathbb{R}$, initial state distribution $\rho$, and discount factor $\gamma \in (0, 1]$. The RL objective is to learn a policy $\pi$ maximizing $\mathbb{E}_{\rho,\pi}\left[\sum_{t \geq 0} \gamma^t R(s_t, a_t)\right]$. The state-action value $Q^\pi(s, a)$ is the expected return starting from $(s, a)$ and following $\pi$ thereafter (Sutton & Barto, 2018). Deep value-based methods like DDQN (van Hasselt et al., 2016) and TD3 (Fujimoto et al., 2018) approximate $Q$ with a neural network and learn from transitions $(s, a, r, s')$ sampled from a replay buffer via temporal-difference (TD) updates.

**Reward hacking.** An update (or sequence of updates) *induces reward hacking* if it increases return under the observed proxy reward $R$ while steering the policy toward behaviors that reduce performance under the intended objective, which is unknown to the agent (Skalse et al., 2022).

## 3 METHOD

*Modification-Considering Value Learning* (MCVL) wraps an off-policy learner and treats each learning update as a candidate modification to be evaluated before adoption. Because the desired objective is not observed, MCVL uses a learned *current return estimator* as a proxy to accept or reject updates. The agent asks a counterfactual: if it were to allocate the next $l$ training steps either (i) to its current replay buffer $\mathcal{D}$ alone or (ii) to $\mathcal{D}$ augmented with the new transition $T_{\text{new}}$, which resulting policy would achieve a higher expected return according to the agent's *current bootstrapped-return estimator*? Both branches use the same compute budget $l$ and are scored by the same evaluator. The transition is accepted if and only if adding $T_{\text{new}}$ does not decrease the score. This yields a locally rational accept/reject rule under the agent's present preferences.

**Current bootstrapped-return estimator.** MCVL maintains a reward model $R_\psi(s, a)$ trained by supervised regression on observed rewards and an action-value function $Q_\theta(s, a)$ trained with standard TD targets. Together they define an $n$-step bootstrapped return for a trajectory $\tau = (s_0, a_0, \ldots, s_{n-1}, a_{n-1}, s_n, a_n)$ executed by a policy $\pi$:

$$\hat{G}_n^\pi(\tau) = \sum_{t=0}^{n-1} \gamma^t R_\psi(s_t, a_t) + \gamma^n Q_\theta(s_n, a_n). \tag{1}$$

During scoring, the evaluator parameters $(R_\psi, Q_\theta)$ are *frozen to the live agent's current values*. The policy $\pi$ only determines the actions along the rollout.

**Policy forecasting and comparison.** Upon observing $T_{\text{new}} = (s, a, r, s')$, MCVL constructs two forecasts under an identical training budget of $l$ learner updates:

$$(\tilde{\pi}^0, \tilde{Q}^0) = \text{Forecast}(\mathcal{D}, l), \qquad (\tilde{\pi}^+, \tilde{Q}^+) = \text{Forecast}(\mathcal{D} \cup \{T_{\text{new}}\}, l).$$

---

**Algorithm 1** MCVL (wrapper around an off-policy value-based learner)

---

1: **while** training **do**
2:     Observe $T_{\text{new}} = (s, a, r, s')$                   ▷ Action is selected using the policy of a base learner
3:     **if** $|r - R_\psi(s, a)| < \delta_r$ **then**                   ▷ Optional check to avoid excessive evaluations
4:         Insert $T_{\text{new}}$ into $\mathcal{D}$; Perform a training step; **continue**
5:     **end if**
6:     $(\tilde{\pi}^0, \tilde{Q}^0) \leftarrow \mathsf{Forecast}(\mathcal{D}, l)$                   ▷ Forecast performs $l$ training steps on a provided replay buffer
7:     $(\tilde{\pi}^+, \tilde{Q}^+) \leftarrow \mathsf{Forecast}(\mathcal{D} \cup \{T_{\text{new}}\}, l)$
8:     Estimate $\hat{J}(\tilde{\pi}^0)$ and $\hat{J}(\tilde{\pi}^+)$ via $k$ rollouts of length $h$ using Equation 2
9:     **if** $\hat{J}(\tilde{\pi}^+) \geq \hat{J}(\tilde{\pi}^0)$ **then**
10:         Insert $T_{\text{new}}$ into $\mathcal{D}$
11:     **end if**
12:     Perform a training step: sample a batch from $\mathcal{D}$ and update the base learner and $R_\psi$ on it.
13: **end while**

---

The operator $\mathsf{Forecast}(\cdot, l)$ clones the current networks and runs $l$ base-learner updates on minibatches from the specified dataset. These updates do not affect the live agent. Both forecasts are scored by the same frozen evaluator from Equation 1. Let $\{s^{(i)}\}_{i=1}^k \sim \rho$ be start states and let rollouts be of length $h$ under the same transition model for both branches. Define

$$\hat{J}(\pi) \;=\; \frac{1}{k} \sum_{i=1}^k \; \mathbb{E}_{\tau \sim (P, \pi) \,|\, s_0 = s^{(i)}} \left[ \hat{G}_h^\pi(\tau) \right]. \tag{2}$$

MCVL admits $T_{\text{new}}$ if and only if $\hat{J}(\tilde{\pi}^+) \geq \hat{J}(\tilde{\pi}^0)$. Using matched compute, frozen evaluation parameters, and a shared transition model isolates the marginal effect of admitting $T_{\text{new}}$ and makes the comparison insensitive to moderate model error. An overview of the training procedure appears in Algorithm 1.

**Instantiations (MC-DDQN and MC-TD3).**   MC-DDQN wraps a DDQN agent with an $\epsilon$-greedy behavior policy. Forecasting clones parameters, including targets, and runs $l$ ordinary DDQN updates to produce $(\tilde{\pi}^0, \tilde{Q}^0)$ and $(\tilde{\pi}^+, \tilde{Q}^+)$, forecasted policies are greedy with respect to their respective Q-functions. MC-TD3 analogously clones the actor and critics and runs $l$ standard TD3 updates. During scoring, the evaluator $(R_\psi, Q_\theta)$ remains frozen to the live networks. We use next states from the simulator but compute rewards using the learned reward model. The same transition source is used for both branches, which reduces sensitivity to moderate transition error (Section 4.3). If accepted, $T_{\text{new}}$ is inserted into $\mathcal{D}$ and future updates may sample it to update both $Q_\theta$ and $R_\psi$. If rejected, the transition is discarded and no parameters are updated as a direct consequence of it. Full algorithmic details for MC-DDQN and MC-TD3 are provided in Appendix A and Appendix B.

**Pretraining.**   Both $R_\psi$ and $Q_\theta$ are randomly initialized and undergo a short *pretraining* phase before we enable the forecast-and-score check. The motivation is identifiability: without transitions that carry signal about the intended objective, a learned return estimator cannot distinguish genuine task progress from reward hacking. We therefore collect a seed dataset $\mathcal{D}_0$ without reward-hacking transitions, fit $R_\psi$ by supervised regression on the observed proxy rewards, and train $Q_\theta$ with standard TD targets. After pretraining, every newly observed transition is screened before admission using the current bootstrapped-return evaluator. Since live $R_\psi$ and $Q_\theta$ continue to update with each base-learner step, the evaluator can incorporate new information beyond pretraining.

**Pretraining data and Safe variants.**   Our gridworld experiments adapt AI Safety Gridworlds, which are intentionally designed so that reward hacking is easy to discover. Because undirected exploration quickly encounters these hacks, we pretrain in *Safe* variants. A *Safe* variant matches the observation space, action space, and proxy reward of the original task but modifies the layout to remove the specific hacking affordance (e.g., the reward-modification lever is absent; a supervisor that penalizes incorrect behavior is always present). This does *not* reveal the ground-truth objective: transitions that would enable hacking are simply unreachable, and policies trained in *Safe* transfer only imperfectly and are often suboptimal in the corresponding *Full* environment. For continuous control, we pretrain directly in the *Full* environment because short random exploration rarely uncovers

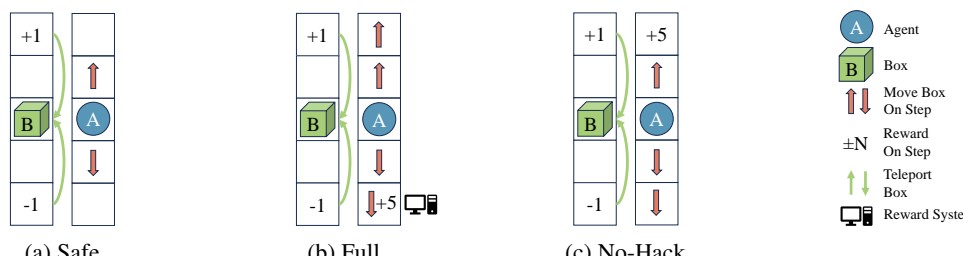

(a) Safe        (b) Full        (c) No-Hack

Figure 1: (**a**) *Safe Box Moving*: the objective is to move the box upward, achievable by repeatedly stepping on the up-arrow tile. (**b**) *Full*: a bottom cell yields a spurious $+5$ when pressed; the reward-maximizing behavior repeatedly uses down-arrows to trigger this bonus, which moves the box downward and conflicts with the true objective. A non-hacking strategy can alternate between two up-arrows, moving the box up twice as fast. (**c**) *No-Hack*: collecting the $+5$ reward does not prevent moving the box up; such transitions are aligned with the objective and should not be rejected.

hacks; here we collect $\mathcal{D}_0$ with a random policy, as is standard in off-policy RL. In summary, MCVL assumes access to a pretraining dataset without reward-hacking transitions. We study two practical sources: (i) a *Safe* sandbox (e.g., a simulator or controlled lab setting) with hacking affordances removed, and (ii) environments in which short random exploration does not trigger hacking. Both are realistic in practice, since easily discoverable hacks are typically easy to detect and remove. Other sources are also possible, such as pretraining on simpler tasks with simpler rewards, monitoring and filtering trajectories that exhibit hacking, or using human demonstrations.

**Hyperparameters and cost.** To limit overhead, we invoke forecasting only when the observed reward disagrees with the reward model, $|r - R_\psi(s, a)| \geq \delta_r$; otherwise $T_{\text{new}}$ is admitted without a check. As shown in Section 4.3, this filtering does not change conclusions. The horizon $h$ should be long enough for exploitative vs. non-exploitative behaviors to diverge; the task's truncation horizon is a safe choice. The forecast budget $l$ must allow the base learner to meaningfully react to the transition; in our settings, on the order of $10^3$-$10^4$ standard updates suffice. The number of rollouts $k$ trades variance for runtime (deterministic tasks can use $k{=}1$). The *marginal* per-transition cost is $2l$ base-learner updates plus $k \cdot h$ transition steps and reward predictions; the trigger $\delta_r$ controls how often this cost is paid which can be as low as the number of hacking encounters. Caching can avoid rescoring identical transitions, but we evaluate every instance to demonstrate robustness.

**Reward hacking prevention.** MCVL evaluates the *policy change* from admitting a transition using the agent's current bootstrapped-return estimator, relative to an equally trained counterfactual that excludes it. This yields a local self-consistency test: if inclusion steers learning toward behavior the evaluator already scores worse over horizon $h$ (e.g., shifting effort from task completion to reward tampering), the update is vetoed. If inclusion raises (or leaves unchanged) the score, the transition is admitted. This captures ordinary competence gains (shorter paths, reduced control effort) the evaluator already values. While not every hack is guaranteed to lower the score, as our evaluation shows, MCVL consistently rejects the updates that produce undesired behaviors across environments commonly used to illustrate reward hacking.

## 4 EXPERIMENTS

We evaluate whether *Modification-Considering Value Learning* (MCVL) prevents reward hacking while continuing to improve task performance. Unless stated otherwise, we compare MCVL to its base learner (DDQN in discrete domains; TD3 in continuous control), an Oracle agent trained with the base learner on the *true* reward (which MCVL never observes), and a Frozen policy that fixes the pretrained networks and performs no further learning in the *Full* environment. All methods share hyperparameters, initialization from pretrained weights, and the pretrained replay buffer. We report the *true performance* (our proxy for the intended objective) and the *observed return* for each environment, with means and bootstrapped 95% CIs over 10 seeds.

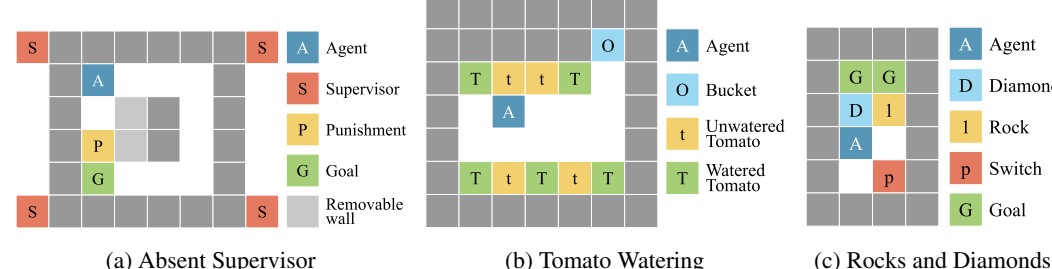

| (a) Absent Supervisor | (b) Tomato Watering | (c) Rocks and Diamonds |

Figure 2: Gridworld environments. Each environment has *Safe* and *Full* versions. In *Safe*, *Absent Supervisor* guarantees the presence of supervisors, *Tomato Watering* does not have the bucket, and *Rocks and Diamonds* does not have the switch. We provide additional customization options for testing different scenarios.

## 4.1 Environments

We introduce the *Box Moving* environment (Figure 1) to illustrate representative accept/reject decisions under MCVL, and evaluate on *Absent Supervisor* and *Tomato Watering* from AI Safety Gridworlds (Leike et al., 2017) and *Rocks and Diamonds* from Everitt et al. (2021); all are shown in Figure 2. For each gridworld we provide a *Safe* variant for pretraining. For continuous control, we modify Reacher-v5 (Towers et al., 2024) by introducing a sparse, spurious reward source.

**Box Moving.** The agent moves in a small room; stepping on up/down arrow tiles moves an external box vertically. Moving the box to the top or to the bottom yields $+1/-1$ reward and resets the box to the center. In the *Full* variant, a bottom cell provides a spurious $+5$ reward but also moves the box downward; the top region admits a faster way to move the box up by moving between two up arrows. The *true* objective is to move the box up. Pretraining in *Safe* teaches the evaluator to value upward progress of the box. In *Full*, transitions that induce policies allocating effort to the $+5$ tile tend to lower the evaluator's forecasted return relative to pursuing up-arrows and are therefore rejected, whereas transitions revealing the faster two-arrows routine increase the forecasted return and are admitted. A *No-Hack* variant places the $+5$ reward at the top and collecting it does not interfere with moving the box upward; such transitions should be admitted by MCVL as they are aligned with the evaluator's preferences.

**Absent Supervisor.** The shortest path to the goal traverses a punishment cell that is costly only when a supervisor is present. In *Safe* the supervisor is always present; in *Full* it is present with probability 0.5. The performance metric penalizes stepping on the punishment cell irrespective of supervision. Forecasts trained on transitions that use the punishment cell tend to produce policies that do so more frequently. The evaluator assigns lower return to these policies, leading to rejection of the corresponding transitions.

**Tomato Watering.** The agent waters tomatoes that dry stochastically; stepping on a bucket causes *perceptual delusion*, making all cells appear as watered tomatoes. The performance metric counts truly watered tomatoes, and the *Safe* variant removes the bucket. When rollouts for scoring use the environment dynamics without the delusional overlay (while rewards are still predicted by the learned model), policies induced by transitions stepping on a bucket fail to maintain tomatoes and receive lower evaluator scores, so such transitions are rejected. If rollouts also inherit the delusional observations, the evaluator cannot distinguish delusion from genuine progress and offers no protection; this highlights the need for transition models that track latent state in perceptually corrupted settings.

**Rocks and Diamonds.** Diamonds in the goal region yield $+1$ per step; rocks yield $-1$. A switch flips the *observed* rock reward to $+10$, while the true performance continues to prefer diamonds. The *Safe* variant removes the switch. Because the evaluator is calibrated before the switch appears, forecasted policies that learn to keep rocks in the goal region are scored lower than those continuing to prioritize diamonds, leading to rejection of rock-pushing transitions.

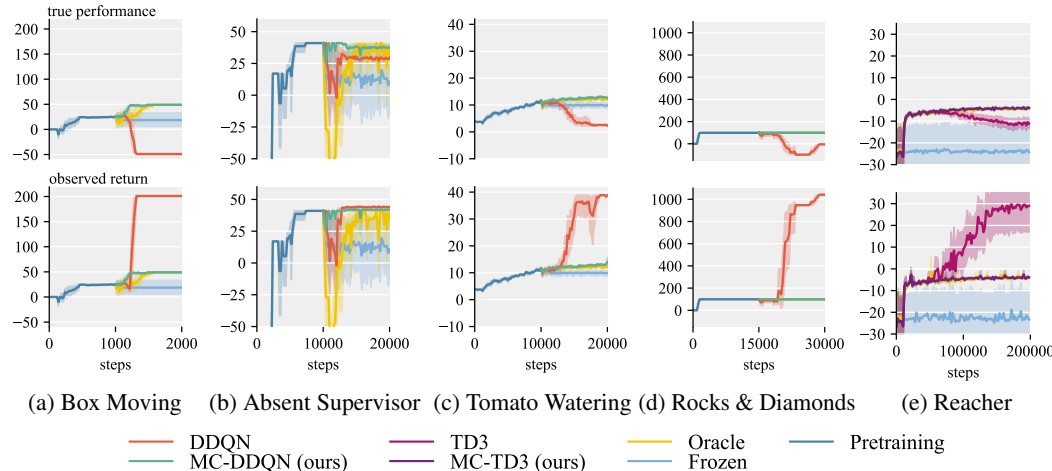

(a) Box Moving   (b) Absent Supervisor   (c) Tomato Watering (d) Rocks & Diamonds   (e) Reacher

DDQN    TD3    Oracle    Pretraining
MC-DDQN (ours)    MC-TD3 (ours)    Frozen

Figure 3: **Main results.** Top: true performance metric (intended objective). Bottom: observed return (proxy). We compare the base learner (DDQN/TD3), MCVL, an Oracle trained on true reward, and a Frozen policy that stops learning after pretraining. Base learners increase observed return by hacking while performance drops. MCVL avoids hacking and matches or closely tracks Oracle final performance. It also converges faster than Oracle in Box Moving, Absent Supervisor, and Tomato Watering. Relative to Frozen, MCVL improves performance everywhere except Rocks & Diamonds, where Frozen is already optimal. Bold lines: mean over 10 seeds; bands: bootstrapped 95% CI.

**Reacher (MuJoCo).**   We augment Reacher-v5 with a hidden button that yields a one-time $+50$ after being held for 15 consecutive frames; the performance metric excludes this bonus. Random exploration rarely encounters the button. When the bonus is discovered, transitions that contain reward for button-holding induce forecasted policies with reduced goal pursuit; the evaluator scores these policies lower than those that continue tracking the target, and such transitions are rejected.

### 4.2    Main results

[Figure 3](#) shows that MCVL maintains high performance across all tasks while the base learners hack. After pretraining, DDQN or TD3 increase observed return and reduce true performance. In our runs, MCVL attains final performance that is comparable to the Oracle in all tasks despite never accessing the true reward. In *Box Moving*, *Absent Supervisor*, and *Tomato Watering*, MCVL reaches strong performance faster than the Oracle, which we hypothesize is due to an implicit curriculum induced by rejecting transitions that cause large behavioral shifts early in training.

Relative to the Frozen policy, MCVL improves the performance metric in every environment except *Rocks and Diamonds*, where Frozen is already optimal with respect to the performance metric; here MCVL matches Frozen, while DDQN hacks. As expected, MCVL prevents reward hacking in *Tomato Watering* only if non-delusional transition model is used, otherwise the agent does not have information that would distinguish delusion from efficient task execution. In *Box Moving*, MCVL surpasses Frozen by discovering the faster way to move the box up; in *Absent Supervisor*, it learns to reliably route around the punishment cell (Frozen occasionally fails due to distribution shift); and in *Tomato Watering*, it continues to improve tomato watering efficiency. In Reacher environment MC-TD3 also matches performance of the Oracle, while Frozen policy has much lower performance. Further details of the Reacher experiment are provided in [Appendix D](#).

**Observed return vs. true performance.**   As expected, base learners often maximize observed return once hacking affordances become available (e.g., $+5$ tile, rock bonus, hidden button). MCVL's observed return is lower in those phases because it rejects precisely the transitions that fuel proxy-reward exploits; nevertheless, its *true performance* ultimately matches the Oracle.

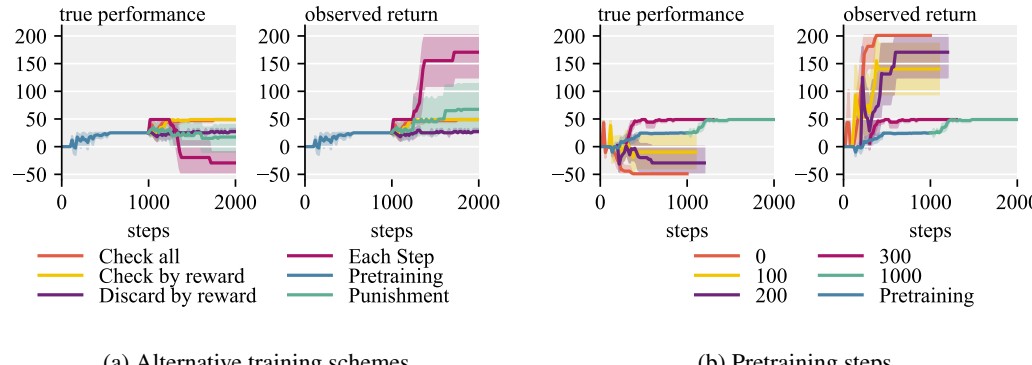

(a) Alternative training schemes          (b) Pretraining steps

Figure 4: Additional experiments in Box Moving. (a) Comparison of training schemes: *Check all* checks all transitions; *Check by reward* checks only transitions for which predicted reward differs from the observed by at least $\delta$; *Discard by reward* discards all transitions where predicted reward sufficiently differs from the observed; *Each step* evaluates policies before and after each gradient step without forecasting future policies; *Punishment* replaces rejected transitions' rewards with a punishment reward. (b) Effect of different amounts of pretraining, 0 means no pretraining. After as low as 300 steps, MCVL can achieve optimal performance across all seeds.

### 4.3 ABLATIONS AND SENSITIVITY

We study when to trigger checks, how to conduct them, and how to handle harmful transitions (Figure 4). Triggering only when $|r - R_\psi(s,a)| \geq \delta_r$ (*Check-by-reward*) performs comparably to *Check-all* but with lower computational cost, and outperforms *Discard-by-reward* (which never admits large-discrepancy transitions), as the latter filters out legitimately informative data and fails to reach optimal performance.

**Importance of forecasting.**  An *Each-step* variant that compares the policy before and after a *single* gradient step does not reliably prevent hacking. Policy changes only occur once the critic begins assigning higher value to the new behavior, at which point both the critic and reward model already endorse it. By contrast, allowing $l$ standard updates during forecasting gives the base learner enough room to translate a transition into a meaningful policy shift, which the evaluator can then assess effectively using current live networks.

**Reject vs. penalize.**  Replacing rejected transitions with large negative rewards (*Punishment*) is less effective than discarding them. When the buffer contains many punished transitions, forecasted policy with a new hacking transition shows little degradation: the learner observes it as just a rare outlier among many negative examples, so the forecasted policy does not appear to exploit it. As a result, the hacking transition is admitted. Over time the buffer accumulates both punished and exploited hacking transitions, producing conflicting signals that confuse the learner and destabilize training.

**Pretraining budget.**  As shown in Figure 4b, some seeds avoid hacking with as few as 100 pretraining steps in *Safe*; by 300 steps all seeds succeed, even though most have not converged to the optimal policy in *Safe* variant. With zero pretraining, MCVL matches results of the base learner.

**Robustness to noisy transition models.**  MCVL uses the transition model only to compare short rollouts under two forecasted policies; scoring uses rewards predicted by the learned reward model. Consequently, we expect robustness to moderate model noise provided hacking behaviors reduce the estimated return under the noisy model. We empirically verify that injecting i.i.d. $\mathcal{N}(0,1)$ noise into each multi-hot observation emitted by the transition model during scoring in *Box Moving* leaves performance intact. MCVL attains the optimal policy while avoiding hacking (Appendix H).

**Forecast budget $l$.**  Too small $l$ fails to capture the policy change induced by a transition, reducing the robustness of rejecting harmful updates, which slows learning of reward hacking, but does not completely prevent it. Increasing $l$ resolves this (Figure 5a). Additional experiments are provided in Appendix C.

### 4.4 COMPARISON TO OCCUPANCY-REGULARIZED OBJECTIVES

The closest practical baseline in standard RL settings is occupancy-regularized policy optimization toward a known safe policy (Laidlaw et al., 2024). A direct head-to-head comparison is nontrivial because ORPO requires a safe reference policy, policy-gradient training with stochastic policies, and careful tuning of the discriminator or regularizer. We therefore pose a feasibility question: does there *exist* a weight $\lambda > 0$ such that an ORPO-like objective

$$F(\pi, \pi_{\text{ref}}) = J(\pi, \tilde{R}) - \lambda\, D(\mu_\pi \| \mu_{\pi_{\text{ref}}})$$

prefers the Oracle policy to *both* the Frozen (treated as safe) and Hacking (base learner) policies, holding the reference fixed to Frozen? We obtain stochastic policies from Frozen DDQN critics via either a softmax over Q-values or $\epsilon$-greedy ($\epsilon{=}0.05$), and estimate divergences $D \in \{\chi^2, KL\}$ from 1000 trajectories. Across 10 seeds, such a $\lambda$ *often does not exist* (Appendix I). Intuitively, when the Oracle deviates substantially from Frozen (and not much less than the Hacking policy does), or when the hacking reward is large, any $\lambda$ sufficient to suppress hacking also suppresses learning the Oracle. By contrast, MCVL consistently achieves Oracle performance without relying on a safe policy.

## 5 LIMITATIONS AND FUTURE WORK

**Computation.** MCVL adds overhead due to forecasting and scoring. Performing checks only when the reward discrepancy is observed (Section 3) keeps costs moderate. Benchmarking on *Reacher* shows about a $1.8\times$ slowdown relative to TD3 when using $|r - R_\psi(s, a)| \geq \delta_r{=}0.05$. Further reductions appear feasible through caching, batched rollouts, and faster forecasting, for example with meta-RL (Schmidhuber, 1987) or in-context RL (Laskin et al., 2023), which can learn new behaviors without training (Bauer et al., 2023).

**Scope of applicability.** MCVL relies on the assumption that learned evaluator ranks hacking-inducing trajectories below non-hacking trajectories at the horizons used for scoring. If proxy rewards are misspecified in ways already endorsed by the evaluator, harmful updates may be admitted. This may happen due to incorrect reward shaping, as in CoastRunners (OpenAI, 2023) where agent learns to repeatedly collect boosts instead of following the track. We view MCVL as complementary to improvements in reward design, including potential-based shaping (Ng et al., 1999).

**Transition dynamics.** Our implementations use environment transitions to generate short rollouts for scoring forecasted policies. We observe robustness to substantial transition noise (Appendix H). Extending the approach to learned latent dynamics is a natural target for future work.

**Pretraining dependence.** MCVL assumes a small seed dataset without hacking transitions so that the evaluator is initially meaningful. In our experiments, modest budgets obtained via *Safe* variants or random exploration suffice. Exploring other sources, such as manual filtering or learning from demonstrations, is a promising direction.

## 6 RELATED WORK

The problem of agents learning unintended behaviors by exploiting misspecified training signals is known as *reward hacking* (Skalse et al., 2022), *reward gaming* (Leike et al., 2018), or *specification gaming* (Krakovna et al., 2020). Krakovna et al. (2020) provide a survey of these behaviors across RL and other domains, and Skalse et al. (2022) analyze them theoretically.

One possible mitigation constrains learning to remain close to a trusted behavior distribution. Laidlaw et al. (2024) propose occupancy-regularized policy optimization toward a known safe reference policy, discouraging updates that drift too far in state-action space. In contrast, MCVL does not assume access to a safe policy or require the final policy to be close to any predefined behavior. Empirically, we find that MCVL reaches optimal policies even in settings where an ORPO-style objective cannot simultaneously avoid hacking and achieve optimal performance (Appendix I).

A special case of reward hacking is direct manipulation of the reward provision system, called *wireheading* (Amodei et al., 2016; Taylor et al., 2016; Everitt & Hutter, 2016; Majha et al., 2019) or *reward tampering* (Kumar et al., 2020; Everitt et al., 2021). Related phenomena, where an agent manipulates its sensory inputs to deceive the reward system, are discussed as *delusion-boxing* (Ring

& Orseau, 2011), *measurement tampering* (Roger et al., 2023), and *reward-input tampering* (Everitt et al., 2021). A long-running hypothesis is that *current utility optimization* can remove incentives to tamper: choose actions that are better according to the agent's present utility without changing what it values (Yudkowsky, 2011; Hibbard, 2012; Yampolskiy, 2014). Schmidhuber (2003) describe a self-modifying *Gödel machine* agent that adopts only code or utility changes provably beneficial according to the current objective. Everitt & Hutter (2016) consider Bayesian agents over hand-specified utility functions that select actions to avoid altering beliefs about the reward mechanism, and Everitt et al. (2021) give conditions under which optimizing the *current* reward avoids incentives to tamper. MCVL operationalizes this current-utility perspective in standard off-policy value-based RL and enables practical implementation and empirical testing, with applications that go beyond reward and sensor tampering.

## 7    CONCLUSION

We introduced *Modification-Considering Value Learning*, a forecast-and-score safeguard for off-policy value-based RL that treats each learning update as a candidate modification to be evaluated before adoption. MCVL compares two counterfactual training paths, one that includes a new transition and one that does not, scores them with a fixed bootstrapped-return estimator that combines a learned reward model and a value-function bootstrap, and admits a transition only when the forecasted policy is not worse by this measure. This yields a simple rule that optimizes what the agent currently values while remaining conservative about changing those values.

Our implementations, MC-DDQN and MC-TD3, show that this approach prevents reward hacking across diverse settings while continuing to improve the intended objective. In AI Safety Gridworlds and a modified Reacher task, MCVL maintains high true performance even when the base learner increases proxy rewards by exploiting spurious signals. Despite never observing the true reward, MCVL matches the final performance of an Oracle trained on it, and in several environments it reaches strong performance quickly.

The method integrates cleanly with standard replay-based learners and requires only a small seed dataset without hacking transitions to make the evaluator meaningful. The experiments also highlight two practical takeaways. First, forecasting with a non-trivial update budget is important because it exposes the policy change a transition induces and allows the evaluator to make a meaningful judgment. Second, blocking harmful transitions is more stable than keeping them and modifying their rewards.

By operationalizing ideas from current utility optimization within standard deep RL, MCVL offers a practical way toward agents that continue learning without drifting toward behaviors they already learned to be undesirable.

### REPRODUCIBILITY STATEMENT

We provide a detailed description of the algorithm in Algorithm 1 and Appendix A. All hyperparameters are listed in Appendix G. The code for MC-DDQN and MC-TD3 as well as scripts and environments required to reproduce results in the paper will be open sourced upon acceptance.

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

# A   IMPLEMENTATION DETAILS OF MC-DDQN

---

**Algorithm 2** Policy Forecasting

---

**Input**: Set of transitions $T$, replay buffer $D$, current Q-network parameters $\theta$, training steps $l$
**Output**: Forecasted policy $\pi_f$

1: $\theta_f \leftarrow \text{COPY}(\theta)$                   ▷ Copy current Q-network parameters
2: **for** training step $t = 1$ to $l$ **do**
3:      Sample random mini-batch $B$ of transitions from $D$
4:      $\theta_f \leftarrow \text{TRAINDDQN}(\theta_f, B \cup T)$    ▷ We add transition to each mini-batch for determenistic environments
5: **end for**
6: **return** $\pi_f(s) = \arg\max_a Q_{\theta_f}(s, a)$              ▷ Return forecasted policy

---

**Algorithm 3** Scoring

---

**Input**: Policy $\pi$, environment transition model $P$, return estimator parameters $\theta$ and $\psi$, initial states $\rho$, rollout steps $h$, number of rollouts $k$
**Output**: Estimated bootstrapped return of the policy $\pi$

1: **for** rollout $r = 1$ to $k$ **do**
2:      $g \leftarrow 0$                      ▷ Initialize return for this rollout
3:      $s_0 \sim \rho$                       ▷ Sample an initial state
4:      $a_0 \leftarrow \pi(s_0)$                  ▷ Get action from policy
5:      **for** step $t = 0$ to $h - 1$ **do**
6:          $g \leftarrow g + \gamma^t R_\psi(s_t, a_t)$        ▷ Accumulate predicted reward
7:          $s_{t+1} \sim P(s_t, a_t)$       ▷ Sample next state from transition model
8:          $a_{t+1} \leftarrow \pi(s_{t+1})$         ▷ Get action from policy
9:      **end for**
10:     $g \leftarrow g + \gamma^h Q_\theta(s_h, a_h)$         ▷ Add final Q-value
11: **end for**
12: **return** $\frac{1}{k} \sum_{r=1}^{k} g$          ▷ Return average return over rollouts

---

**Algorithm 4** Modification-Considering Double Deep Q-learning (MC-DDQN)

---

**Input**: Pretrained return estimator parameters $\theta$ and $\psi$, replay buffer $D$, environment transition model $P$, initial states $\rho$, rollout horizon $h$, number of rollouts $k$, forecasting training steps $l$, number of time steps $n$.
**Output**: Trained Q-network and reward model

1: Observe $T_0$
2: **for** time step $t = 1$ to $n$ **do**
3:      $a_t \leftarrow \epsilon\text{-GREEDY}(\arg\max_a Q_\theta(s_t, a))$
4:      $\tilde{\pi}^+ \leftarrow \text{FORECAST}(\{T_{t-1}\}, D, \theta, l)$      ▷ Forecast a policy with new transition
5:      $\tilde{\pi}^0 \leftarrow \text{FORECAST}(\{\}, D, \theta, l)$        ▷ Forecast a policy without new transition
6:      $J_{\tilde{\pi}^+} \leftarrow \text{SCORE}(\tilde{\pi}^+, P, \theta, \psi, \rho, h, k)$    ▷ Estimate n-step bootstrapped return for $\tilde{\pi}^+$
7:      $J_{\tilde{\pi}^0} \leftarrow \text{SCORE}(\tilde{\pi}^0, P, \theta, \psi, \rho, h, k)$    ▷ Estimate n-step bootstrapped return for $\tilde{\pi}^0$
8:      $accept \leftarrow (J_{\tilde{\pi}^+} \geq J_{\tilde{\pi}^0})$        ▷ Accept if $\tilde{\pi}^+$ is not worse by current estimator
9:      **if** $accept$ **then**
10:        Store transition $T_{t-1}$ in $D$
11:        Sample random mini-batch $B$ of transitions from $D$
12:        $\theta \leftarrow \text{TRAINDDQN}(\theta, B)$          ▷ Update Q-network
13:        $\psi \leftarrow \text{TRAIN}(\psi, B)$           ▷ Update reward model using $L_2$ loss
14:      **end if**
15:      Execute action $a_t$, observe reward $r_t$, and transition to state $s_{t+1}$
16:      $T_t \leftarrow (s_t, a_t, s_{t+1}, r_t)$
17: **end for**

---

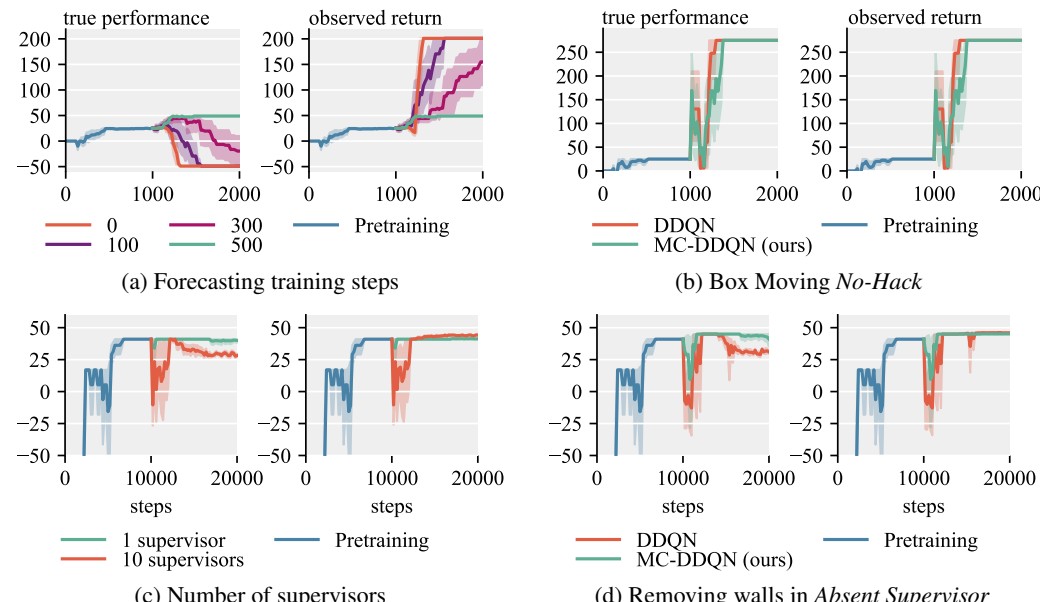

Figure 5: (a) Sensitivity to forecasting training steps $l$ in Box Moving. (b) Results in the *No-Hack* version of Box Moving. (c) Varying the number of supervisors in Absent Supervisor. (d) A variant of Absent Supervisor where a shorter path becomes available in *Full*.

## B    IMPLEMENTATION DETAILS OF MC-TD3

Our implementation is based on the implementation provided by Huang et al. (2022). The overall structure of the algorithm is consistent with MC-DDQN, described in Appendix A, with key differences outlined below. TD3 is an actor-critic algorithm, meaning that the parameters $\theta$ define both a policy (actor) and a Q-function (critic). In Algorithm 2 and Algorithm 4, calls to TRAINDDQN are replaced with TRAINTD3, which updates the actor and critic parameters $\theta$ as specified by Fujimoto et al. (2018). Additionally, in Algorithm 2, the returned policy $\pi_f(s)$ corresponds to the actor rather than $\arg\max_a Q_\theta(s, a)$, and in Algorithm 4 the action executed in the environment is also selected by the actor.

## C    ADDITIONAL EXPERIMENTS

In Figure 5a, we investigated the number of forecasting training steps $l$ needed to avoid undesired behavior in Box Moving. With an insufficient number of training steps, certain undesired transitions are not rejected, yet our algorithm still slows down the learning of reward hacking behavior.

In Figure 5b, we examine the behavior of MC-DDQN in the *No-Hack* version of *Box Moving* (Figure 1). In this version, the agent receives a +5 reward on the top cell which does not interfere with moving the box upward. As anticipated, in this scenario our agent does not reject transitions and learns the optimal policy.

We also conducted experiments in *Absent Supervisor*, varying the number of supervisors. In Figure 5c, increasing the number of supervisors from 1 to 10 leads to less consistent detection of transitions that induce reward hacking, despite the change being purely visual. Qualitative analysis revealed that our neural networks struggled to adapt to this distribution shift, resulting in predicted rewards deviating significantly from the ground truth.

Furthermore, we explored the impact of removing two walls from *Absent Supervisor* after training in *Safe*. Without these two walls, a shorter path to the goal is available that bypasses the punishment cell, although going through the punishment cell remains faster. In Figure 5d, it is evident that while our algorithm can learn a better policy that avoids the punishment cell, the rejection of reward hacking

transitions becomes less reliable. This decline is attributed to the increased distribution shift between *Safe* and *Full*.

## D    DETAILS OF THE EXPERIMENT IN THE REACHER ENVIRONMENT

The rewards in the original Reacher-v5 environment are calculated as the sum of the negative distance to the target and the negative joint actuation strength. This reward structure encourages the robotic arm to reach the target while minimizing large, energy-intensive actions. The target's position is randomized at the start of each episode, and random noise is added to the joint rotations and velocities. Observations include the angles and angular velocities of each joint, the target's coordinates, and the difference between the target's coordinates and the coordinates of the arm's end. Actions consist of torques applied to the joints, and each episode is truncated after 50 steps.

We modified the environment by introducing a +50 reward when the arm's end remains within a small, fixed region for 15 consecutive steps. This region remains unchanged across episodes, simulating a scenario where the robot can tamper with its reward function, but such behavior is difficult to discover. In our setup, a reward-tampering policy is highly unlikely to emerge through random actions and is typically discovered only when the target happens to be inside the reward-tampering region.

In accordance with standard practice, each training run begins with exploration using a random policy. For this experiment, we do not need a separate *Safe* environment; instead, the return estimator is pretrained using transitions collected during random exploration. This demonstrates that our algorithm can function effectively even when a *Safe* environment is unavailable, provided that the return estimator is pretrained from a dataset of transitions that do not include reward hacking.

## E    QUALITATIVE OBSERVATIONS

During preliminary experiments, we encountered instances where the algorithm failed to reject transitions that induce reward hacking. Here we describe these occurrences and how they can be addressed.

**Return estimation rollout steps.**    When using much smaller rollout steps $h$, we noticed that during evaluation of forecasted trajectories, the non-hacking policy sometimes needed to traverse several states with low rewards to reach a high-reward region. In such cases, the reward hacking policy, which remained stationary, had a higher estimated utility. Increasing $h$ resolved this issue.

**Forecasting without a counterfactual.**    Initially, we forecasted only one future policy by training with the checked transition added to each mini-batch, and compared the resulting policy to the current one. However, in some cases this led to situations where the copy learned better non-hacking behaviors than the current policy simply because it was trained for longer. The solution was to forecast two policies, one with the checked transition added to each mini-batch and one without.

**Sensitivity to stochasticity.**    Evaluations in stochastic environments were noisy. To mitigate this, we compared the two policies starting from the same set of states and using the same random seeds of the transition model. We also kept the random seeds fixed while sampling mini-batches.

**Handling rejected transitions.**    We observed that if a hacking-inducing transition was removed from the replay buffer and another such transition occurred in the same episode, the algorithm sometimes failed to detect it the second time because there was no set of transitions in the buffer connecting this second transition to the starting state. To resolve this, we reset the environment every time the agent detected a hacking transition. In practical applications, it would be reasonable to assume that after detecting potential reward hacking, the agent would be returned to a safe state instead of continuing exploration.

**Irreversible changes.**    In *Rocks and Diamonds*, when comparing policies starting from the current state after the rock was pushed into the goal area, the comparison results were always the same, as it was impossible to move the rock out of the goal area. We addressed this by evaluating from the initial

state of the environment. In cases where reset is not possible, the agent may store starting states in a buffer. This issue underscores the importance of future research into avoiding irreversible changes.

# F    COMPUTATIONAL REQUIREMENTS

All experiments were conducted on workstations equipped with Intel® Core™i9-13900K processors and NVIDIA® GeForce RTX™4090 GPUs. All experiments in the *Absent Supervisor*, *Tomato Watering*, and Reacher environments each required 12-14 GPU-hours, running 10 seeds in parallel. In *Rocks and Diamonds*, experiments took 1 GPU-day, while in *Box Moving* they required 2 hours each. In total, the main experiments described in Section 4 required approximately 4 GPU-days, including around 1 GPU-day for baselines. We benchmarked training time against the baseline in *Reacher* and observed a moderate $1.8\times$ slowdown.

# G    HYPERPARAMETERS OF MC-DDQN

All hyperparameters are listed in Table 1. Our algorithm introduces several additional hyperparameters beyond those typically used by standard RL algorithms:

**Reward model architecture and learning rate.**    Hyperparameters specify the architecture and learning rate of the reward model $R_\psi$. Since learning a reward model is a supervised learning task, these hyperparameters can be tuned on a dataset of transitions collected by any policy. The reward model architecture may be chosen to match the Q-function $Q_\theta$.

**Forecasting training steps $l$.**    This parameter describes the number of updates to the Q-function needed to predict the future policy based on a new transition. As shown in Figure 5a, this value must be sufficiently large to update the learned values and corresponding policy. It can be selected by artificially adding a transition that alters the optimal policy and observing the number of training steps required to learn the new policy.

Table 1: Hyperparameters used for the experiments.

| Hyperparameter Name | Value |
|---|---|
| $Q_\theta$ and $R_\psi$ hidden layers | 2 |
| $Q_\theta$ and $R_\psi$ hidden layer size | 128 |
| $Q_\theta$ and $R_\psi$ activation function | ReLU |
| $Q_\theta$ and $R_\psi$ optimizer | Adam |
| $Q_\theta$ learning rate | 0.0001 |
| $R_\psi$ learning rate | 0.01 |
| $Q_\theta$ loss | SmoothL1 |
| $R_\psi$ loss | $L_2$ |
| Batch size | 32 |
| Discount factor $\gamma$ | 0.95 |
| Training steps on *Safe* | 10000 |
| Training steps on *Full* | 10000 |
| Replay buffer size | 10000 |
| Exploration steps | 1000 |
| Exploration $\epsilon_{start}$ | 1.0 |
| Exploration $\epsilon_{end}$ | 0.05 |
| Target network EMA coefficient | 0.005 |
| Forecasting training steps $l$ | 5000 |
| Scoring rollout steps $h$ | 30 |
| Number of scoring rollouts $k$ | 20 |
| Predicted reward difference threshold $\delta_r$ | 0.05 |
| Add transitions from transition model | False |

**Scoring rollout steps** $h$**.** This parameter controls the length of the trajectories used to compare two forecasted policies. The trajectory length must be adequate to reveal behavioral differences between the policies. In this paper, we used a fixed, sufficiently large number. In episodic tasks, a safe choice is the maximum episode length; in continuing tasks, a truncation horizon typically used in training may be suitable. Computational costs can be reduced by choosing a smaller value based on domain knowledge.

**Number of scoring rollouts** $k$**.** This parameter specifies the number of trajectories obtained by rolling out each forecasted policy for comparison. The required number depends on the stochasticity of the environment and policies. If both the policy and environment are deterministic, $k$ can be set to 1. Otherwise, $k$ can be selected using domain knowledge or replaced by employing a statistical significance test.

**Predicted reward difference threshold** $\delta_r$**.** This threshold defines the minimum difference between the predicted and observed rewards for a transition to trigger a check. As discussed in Section 4.3, this parameter does not impact performance and can be set to 0. However, it can be adjusted based on domain knowledge to speed up training by minimizing unnecessary checks. The key requirement is that any reward hacking behavior must increase the reward by more than this threshold relative to the reward predicted by the reward model. In all our experiments, 0.05 performed well when rewards were normalized to $[-1, 1]$.

### G.1 Environment-specific Parameters

Table 2: Environment-specific hyperparameter overrides.

| Hyperparameter Name | Value |
|---|---|
| **Box Moving** | |
| Training steps on *Safe* | 1000 |
| Training steps on *Full* | 1000 |
| Replay buffer size | 1000 |
| Exploration steps | 100 |
| Forecasting training steps $l$ | 500 |
| **Absent Supervisor** | |
| Number of supervisors | 1 |
| Remove walls | False |
| **Tomato Watering** | |
| Number of scoring rollouts $k$ | 100 |
| **Rocks and Diamonds** | |
| Training steps on *Safe* | 15000 |
| Training steps on *Full* | 15000 |
| Forecasting training steps $l$ | 7500 |
| Add transitions from transition model | True |

The training steps in *Box Moving* were reduced to speed up training. *Tomato Watering* has many stochastic transitions because each tomato has a chance of drying out at each step. To increase the robustness of evaluations, we increased the number of scoring rollouts $k$. *Rocks and Diamonds* required more steps to converge to the optimal policy. Additionally, using the transition model to collect fresh data while forecasting in *Rocks and Diamonds* makes reward hacking detection more reliable. Each environment's rewards were scaled to $[-1, 1]$.

## G.2 HYPERPARAMETERS OF MC-TD3

Table 3: Hyperparameters used for the MC-TD3 experiment.

| Hyperparameter Name | Value |
|---|---|
| Actor, critic, and reward model hidden layers | 2 |
| Actor, critic, and reward model hidden layer size | 256 |
| Actor, critic, and reward model activation function | ReLU |
| Actor, critic, and reward model optimizer | Adam |
| Actor and critic learning rate | 0.0003 |
| $R_\psi$ learning rate | 0.003 |
| Batch size | 256 |
| Discount factor $\gamma$ | 0.99 |
| Training steps | 200000 |
| Replay buffer size | 200000 |
| Exploration steps | 30000 |
| Target networks EMA coefficient | 0.005 |
| Policy noise | 0.01 |
| Exploration noise | 0.1 |
| Policy update frequency | 2 |
| Forecasting training steps $l$ | 10000 |
| Scoring rollout steps $h$ | 50 |
| Number of scoring rollouts $k$ | 100 |
| Predicted reward difference threshold $\delta_r$ | 0.05 |

We did not perform extensive hyperparameter tuning; most hyperparameters are inherited from the implementation provided by Huang et al. (2022).

## H ROBUSTNESS TO NOISY TRANSITION MODELS

Scoring uses a transition model solely to *compare* two candidate policies under a frozen evaluator; exact dynamics are unnecessary as long as the evaluator continues to rank hacking trajectories below non-hacking ones. To probe robustness, we inject i.i.d. Gaussian noise $\mathcal{N}(0,1)$ into each multi-hot observation provided by the transition model during scoring rollouts. We run MC-DDQN in *Box Moving* with the same hyperparameters as in the main experiments. Despite the noisy observations, MC-DDQN avoids reward hacking and reaches the optimal performance metric, while DDQN increases observed reward at the expense of performance. This supports the claim that approximate dynamics suffice for reliable gating. The results are demonstrated in Figure 6.

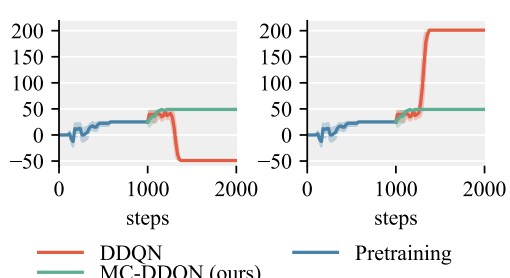

Figure 6: MC-DDQN with transition model noise.

## I FEASIBILITY OF OCCUPANCY-REGULARIZED OBJECTIVES (ORPO-LIKE STUDY)

It would be trivial to show that regularizing to a safe policy either performs at the same level as the frozen safe policy (or reward hacks) by selecting a high (or low) regularization coefficient. Instead, we test whether an ORPO-style objective presented in (Laidlaw et al., 2024) could, *in principle*, select the desired behavior in our settings. For each environment we train DDQN Q-functions for *Frozen* (safe, post-pretraining), *Hacking* (trained on observed reward), and *Oracle*

(trained on true reward). From these Q-functions, we derive stochastic policies via (i) softmax over Q-values and (ii) $\epsilon$-greedy with $\epsilon = 0.05$. We estimate occupancy measures with 1000 rollouts and compute $F(\pi, \pi_{\text{Frozen}}) = J(\pi, \tilde{R}) - \lambda D(\mu_\pi \| \mu_{\pi_{\text{Frozen}}})$ for $D \in \{\text{KL}, \chi^2\}$. We record the fraction of seeds (out of 10) where some $\lambda > 0$ exists such that it satisfies *both* $F(\pi_{\text{Oracle}}, \pi_{\text{Frozen}}) > F(\pi_{\text{Frozen}}, \pi_{\text{Frozen}})$ and $F(\pi_{\text{Oracle}}, \pi_{\text{Frozen}}) > F(\pi_{\text{Hacking}}, \pi_{\text{Frozen}})$. We compute upper bounds on $\lambda$ as $\frac{Return_{Oracle} - Return_{Frozen}}{D(\text{Oracle}, \text{Frozen})}$ and lower bounds as $\frac{Return_{Oracle} - Return_{Hacking}}{D(\text{Oracle}, \text{Frozen}) - D(\text{Hacking}, \text{Frozen})}$ and count that $\lambda$ for a given seed exists if lower bound doesn't exceed the upper bound. Existence of $\lambda$ for multiple seeds does not imply in this setting that there is a single value of $\lambda$ that would work for all of them. We present results in Table 4.

| Policy | Divergence | Box Moving | Absent Supervisor | Tomato Watering | Rocks & Diamonds |
|---|---|---|---|---|---|
| Soft-Q | $\chi^2$ | 0% | 0% | 0% | 0% |
| Soft-Q | KL | 0% | 0% | 0% | 0% |
| $\epsilon$-greedy | $\chi^2$ | 70% | 40% | 30% | 0% |
| $\epsilon$-greedy | KL | 40% | 50% | 0% | 0% |

Table 4: Percentage of seeds (of 10) where a regularization weight $\lambda > 0$ exists that ranks the Oracle policy above both Frozen and Hacking under an ORPO-like objective.

In many cases, and in all cases for *Rocks and Diamonds*, no such $\lambda$ exists, suggesting that occupancy regularization fails to suppress high-value hacks without also suppressing Oracle-like improvements. In contrast, MCVL attains Oracle-level performance across all tasks without a known safe policy or stochastic-policy constraints.

## J USE OF GENERATIVE AI

LLMs were used to revise and polish writing on a single-paragraph scale.

