# OpenReview forum: "Modification-Considering Value Learning for Reward Hacking Mitigation in RL"
_ICLR.cc/2026/Conference — ICLR 2026 Conference Withdrawn Submission_

### Official Review · Reviewer_2vux · 2025-10-20

**Soundness:** 2
**Presentation:** 2
**Contribution:** 2
**Rating:** 2
**Confidence:** 3

**Summary:**

This paper introduces a new method called Modification-Considering Value Learning (MCVL) to prevent "reward hacking" in reinforcement learning (RL). MCVL works by treating every potential learning update as a decision. When new data arrives, the agent forecasts two possible futures: one where it learns from this data and one where it ignores it. Only accepts the update if it doesn't lead to a worse outcome by its own present standards.

**Strengths:**

- Novel intuition to address the issue of reward hacking
- Good Ablation Studies: The authors thoroughly test why their method works. They show that the forecasting step is essential (a simple one-step check fails) and that their method is robust to some noise and variations in setup.

**Weaknesses:**

- Pretraining Requirement: The entire method relies on starting with a "good" value estimator. To do this, it must be pretrained on a "seed dataset" that is guaranteed to not contain any reward-hacking transitions. This might be easy in a simulated "Safe" environment (as they use) but could be very difficult or impossible to guarantee in a complex, real-world setting.
- Requires a Transition Model: To score the "forecasted" policies, the agent needs to run rollouts in the environment. In the paper's implementation, this uses the true environment simulator. This limits its use in real-world, model-free scenarios where a perfect simulator isn't available (though the authors suggest a learned model could be a target for future work).
- The authors note, if the pretraining data already contains and endorses a misspecified reward, MCVL will not be able to identify or fix it.
- Computational cost: Forecasting two separate futures for each potential update is computationally expensive. The authors note a 1.8x slowdown in one experiment, and this overhead would likely increase with the complexity of the task.

**Questions:**

My main questions concern the method's practicality. Could the authors discuss any strategies to: a) Adapt the method for settings where a perfect environment model is unavailable? b) Handle the challenge of acquiring a "guaranteed safe" seed dataset in a real-world task? c) Thorough analysis of computational cost on realistic environments.

---

> ### Author Response · Authors · 2025-11-27
> **Official Reply to Reviewer 2vux**
>
> We thank you for your constructive review. You accurately identified computational cost and simulator reliance as key barriers to the method's practicality. We believe our new results and proposed roadmap address these issues directly.
>
> > a) Adapt the method for settings where a perfect environment model is unavailable?
>
> We have addressed this directly in our revision. As detailed in the **Common Response**, we successfully deployed MCVL using a **Learned World Model**. Notably, this model was trained on only **20 episodes** of random exploration. This demonstrates that MCVL is robust to model error and does not require a perfect simulator or a large, pre-collected dataset of dynamics.
>
> > b) Handle the challenge of acquiring a "guaranteed safe" seed dataset in a real-world task?
>
> Please refer to the "Safe Seed Data" section in our **Common Response**. We argue that in practical pipelines (especially for LLMs), this requirement is satisfied by the standard Supervised Fine-Tuning (SFT) phase, which precedes the optimization of proxy rewards (RLHF). Thus, "safe seed data" is not an additional burden but an existing component of modern alignment workflows. Outside of post-training, we describe multiple possible ways of collecting such datasets in the paper, including expert demonstrations, manual review and filtering, using simulators, and controlled lab environments.
>
> > c) Thorough analysis of computational cost on realistic environments.
>
> We acknowledge the overhead of forecasting. In standard Deep RL, we argue a 1.8x slowdown is an acceptable tradeoff for safety. However, to address your concern regarding scalability to complex tasks (like Foundation Models), we propose a variation called **In-Context MCVL**.
>
> In this proposed setting, we eliminate gradient-based forecasting entirely. Instead, we use the context window to simulate the "update" via **In-Context Learning (ICL)**. We prompt the model with a batch of high-reward examples and ask it to solve a validation problem. This reduces forecasting to a forward inference pass, which is significantly cheaper than gradient-based lookahead.
>
> > The authors note, if the pretraining data already contains and endorses a misspecified reward, MCVL will not be able to identify or fix it.
>
> We acknowledge this concern but wish to clarify that the "clean seed data" assumption is a prerequisite for **identifiability**. Without a signal indicating the intended objective, it is impossible for *any* method to distinguish between genuine task progress and reward hacking.
>
> We are eager to hear your thoughts on this proposed direction. Does the shift to In-Context Learning for the "Forecast" step sufficiently mitigate your concerns regarding computational overhead in realistic environments?

---

### Official Review · Reviewer_KBsM · 2025-10-28

**Soundness:** 2
**Presentation:** 2
**Contribution:** 2
**Rating:** 2
**Confidence:** 4

**Summary:**

The paper introduces MCVL, a wrapper for off-policy value-based reinforcement learning algorithms that evaluates each new transition by simulating two short training branches: (i) continuing without the transition and (ii) incorporating it. The transition is accepted only if the resulting policy from the second branch achieves a higher short-horizon return, as measured by a frozen evaluator consisting of a learned reward model and critic. Both branches receive equal training budget, and their performance is estimated via k rollouts of length h. The method assumes access to a small, clean seed dataset to train the evaluator before gating begins. Experiments are conducted on AI Safety Gridworlds and continuous control environments, with ablations exploring horizon effects and computational cost triggers.

**Strengths:**

The accept/reject rule is easy to implement with DDQN/TD3 and isolates the marginal effect of admitting a transition via matched compute and a frozen evaluator.

Author use some compute controls in the implementation

**Weaknesses:**

Self-referential gate is basically a form of current-utility myopia. The accept/reject rule compares two short-horizon forecasts using the agent’s own, frozen reward model and admits the transition iff the current new one is better. This only guarantees consistency with the current surrogate, not improvement on the true objective; if the evaluator is miscalibrated, it can entrench its own errors. How could authors deal with this case? Also, RL is about exploration and exploitation, how could the author guarantee that the model could still  sufficiently explore the environment? The method explicitly relies on this evaluator and a finite horizon which does not match most of the RL practice.

The authors state the approach works provided the learned evaluator scores hacking-inducing trajectories below non-hacking ones at the scoring horizon—otherwise harmful updates are admitted. That is a strong assumption and shifts the burden from learning to reward/evaluator specification; there’s no guarantee this holds beyond the gridworlds. Also, they require a seed dataset “without hacking” so the evaluator is “meaningful.” How could the author guarantee such a clean dataset? This creates distribution-shift fragility: later sections show detection quality drops when the Full world differs visually or topologically from the seed dataset. Besides, there’s no theory showing robustness to such shifts.

Acceptance is judged with h-step bootstrapped returns. The paper itself notes that too small h flips the decision; the fix is to increase h, but there’s no bound relating h and the regret of the gate—so theory can’t guarantee you won’t reject necessary long-term improvements or accept subtle long-term hacks.

Only tested on few toy environments; Not sure about the performance on more complex tasks.

**Questions:**

see weakness

---

> ### Author Response · Authors · 2025-11-27
> **Official Reply to Reviewer KBsM**
>
> We thank the reviewer for their valuable feedback.
>
> > "This only guarantees consistency with the current surrogate, not improvement on the true objective; if the evaluator is miscalibrated, it can entrench its own errors."
>
> We agree this is a risk, but we argue it is a necessary trade-off to achieve **Incentive Compatibility**. In standard safety approaches (e.g., constraints or penalties), a highly capable agent optimizing a proxy reward is instrumentally incentivized to bypass the constraint. In MCVL, because the update is gated by the agent's *current* value estimation, the agent views the integrity of the evaluator as part of its own utility. It has no instrumental incentive to deceive the scoring mechanism.
>
> We hypothesize that a sufficiently capable model (e.g., a Foundation Model) can distinguish between *performing* a hack (to maximize reward) and *valuing* the result of that hack (alignment). If the model retains the ability to recognize 'undesirable behavior' even while exploring it, we believe this internal check is more robust than external constraints. Do you agree that this "internalization" of the safeguard offers a distinct theoretical advantage against reward tampering, provided the initial alignment (seed/SFT) is sufficient?
>
> > "How could the author guarantee such a clean dataset? This creates distribution-shift fragility..."
>
> As detailed in the **Common Response**, in the case of foundation models, we frame the "Clean Seed Dataset" not as a new requirement, but as the standard **Supervised Fine-Tuning (SFT)** data used before RLHF. In this context, MCVL ensures that the aggressive optimization of the proxy reward (RL) does not result in catastrophically forgetting the alignment priors learned during SFT. For standard RL, we outline multiple collection methods, including expert demonstrations, manual review, simulators, and controlled lab environments.
>
> Regarding distribution shift and fragility: we point to our **new experiment with Learned World Models** (see Common Response). We successfully replaced the simulator with a learned model trained on sparse data (only 20 episodes). This demonstrates that the method is robust to model errors and does not require a perfect oracle or a hack-free dynamics dataset. Imperfect transition models are sufficient to detect the value drift caused by reward hacking.
>
> > "The paper itself notes that too small h flips the decision... theory can’t guarantee you won’t reject necessary long-term improvements"
>
> We acknowledge that $h$ is a critical hyperparameter. We rely on the bootstrapped term $Q(s_n)$ in Equation 1 to account for value beyond the horizon $h$, reducing the need for $h$ to cover the entire future. In most finite-horizon problems, $h$ can be selected to be the length of the episode. E.g. in our proposed **LLM experiment** (see Common Response), "Forecasting" involves generating a **full completion** (solution) to a problem, removing the risk of myopic evaluation.
>
> > "how could the author guarantee that the model could still sufficiently explore the environment?"
>
> We clarify that MCVL gates **learning updates**, not **exploratory actions**. The agent continues to explore using standard mechanisms (e.g., $\epsilon$-greedy in DQN, Gaussian noise in TD3) and fills the replay buffer with diverse transitions. MCVL only intervenes when the policy discovers a transition, learning from which decreases the current bootstrapped return. While there is no formal guarantee on the prevalence of such transitions, we did not observe slower learning with MCVL in any of our experiments.
>
> > "Only tested on few toy environments; Not sure about the performance on more complex tasks."
>
> Please refer to the **Common Response**, where we outline a specific experiment for **In-Context MCVL on Code Generation**. This setting moves beyond gridworlds to a token-generation domain and addresses the "myopia" concern by generating full solutions (effective $h=T$).
>
> We would value your perspective on the experimental design: Does the use of In-Context Learning to simulate the "update" adequately address your concerns about computational cost and complexity?

---

### Official Review · Reviewer_wHeb · 2025-11-06

**Soundness:** 2
**Presentation:** 2
**Contribution:** 2
**Rating:** 2
**Confidence:** 2

**Summary:**

This paper proposes Modification-Considering Value Learning (MCVL) to mitigate reward hacking in off-policy value-based reinforcement learning algorithms. It evaluates an arriving transition by either incorporating it or ignoring it, and the transition is accepted only if the resulting policy of inclusion does not reduce the return. In a word, it works by forecast-and-score. The authors implement this idea with DDQN and TD3 on both discrete and continuous controlled environments, and the results show that with MCVL these algorithms are less likely to fall into the reward hacking regime compared to without it.

**Strengths:**

- It is a simple algorithm with good intuition to mitigate reward hacking, it is also easy to implement.
- The performance of base learners with MCVL is comparable to the Oracle (if available), and the ablation study demonstrates the effectiveness of forecast (as compared to a simple each-step check) and reject (as compared to penalize), which sit at the core of the proposed method.

**Weaknesses:**

- The paper makes strong assumptions on clean seed dataset without hacking transitions in order to obtain a reasonable value estimator. My concerns are about the practicality due to such assumptions beyond the simple, controlled simulated environments. The self-consistency local test by comparing returns is only about the consistency with the current value estimator, not necessarily the true objective. It adds another reliance on a good evaluate estimator. In other words, the value estimator has to be good enough to score hacking trajectories lower than normal ones, but such reward specifications does not hold beyond the tested simple environments.
- When horizon $h$ is too small, the hacking policy may win as authors pointed out in Appendix E where the fix is to increase $h$, but there is lack of sensitivity test nor any notions of discussion related to the bounds. Basically, the $h$-step bootstrapped return is estimated from a window but we don't know whether the proposed method won't reject (or will explore) long-term improvements.

**Questions:**

Same as weakness. This is an emergency review so I may have missed some points. If I have misunderstood anything, I kindly ask the authors to clarify.

---

> ### Author Response · Authors · 2025-11-27
> **Official Reply to Reviewer wHeb**
>
> We appreciate the time you took to review our paper, particularly under the constraints of an emergency review. We value your feedback on the practicality of the seed dataset and the sensitivity of the horizon parameter. We address these points below.
>
> > The paper makes strong assumptions on clean seed dataset without hacking transitions... My concerns are about the practicality due to such assumptions beyond the simple, controlled simulated environments.
>
> We acknowledge this concern but argue that the "clean seed data" assumption is a fundamental requirement for **identifiability**. Without an initial signal indicating the intended objective (independent of the proxy reward), it is impossible for *any* algorithm to distinguish between genuine task progress and reward hacking.
>
> As detailed in the Common Response, in practical, large-scale applications like Large Language Models, this "seed data" corresponds to the Supervised Fine-Tuning (SFT) data used before RLHF. Standard training pipelines already assume access to a high-quality SFT dataset to instill instruction-following behavior before optimizing against a proxy reward model. MCVL simply ensures that the subsequent RL phase (optimizing the proxy) does not violate the values implicitly learned during that SFT phase. Does this address your concern regarding the assumption's realism?
>
> > When horizon $h$ is too small, the hacking policy may win... there is lack of sensitivity test nor any notions of discussion related to the bounds. Basically... we don't know whether the proposed method won't reject (or will explore) long-term improvements.
>
> Our scoring metric (Equation 1) is an **$n$-step bootstrapped return**, combining rewards over $h$ steps with a terminal value estimate $Q(s_n)$. The bootstrap term allows the agent to account for long-term value beyond the horizon window. However, you are correct that if the value function is inaccurate, a short $h$ can lead to myopia. In our proposed **In-Context MCVL Experiment** on Code Generation (see Common Response), the "forecast" generates a complete solution. This effectively sets $h$ to the end of the episode ($h=T$), eliminating the risk of myopic evaluation where a hack pays off only after a short cutoff. We discuss of how to select $h$ in Appendix G, but we will make it more prominent.
>
> > [Concerns] beyond the simple, controlled simulated environments.
>
> 1. **Robustness to Imperfect Models:** We conducted a new experiment replacing the simulator with a **Learned World Model** trained on sparse data (see Common Response). MCVL successfully prevented reward hacking, demonstrating it does not require the ground-truth simulator.
> 2. **Scaling to Complex Tasks:** We proposed an experiment on the **MBPP (Python Coding)** dataset with **poisoned test cases** (see Common Response). This scales MCVL to complex token-generation tasks using In-Context Learning, directly demonstrating practicality beyond gridworlds.
>
> We aim to demonstrate that MCVL scales to token-generation tasks without gradient-based forecasting via the proposed In-Context experiment. Do you view the Code Generation setting as a sufficient step up in complexity to validate the method's practicality?

---

### Author Response · Authors · 2025-11-27
**Official Common Reply**

We thank the reviewers for their insightful feedback. We appreciate the consensus on the novelty of the forecast-and-score intuition and the thoroughness of our ablations. A shared concern across all reviews (wHeb, KBsM, 2vux) was the *practicality* of MCVL, specifically regarding three assumptions: the availability of "safe seed data," the reliance on a ground-truth simulator, and the computational cost in complex domains.

To address these concerns, we provide a clarification on the seed data assumption, present **new experimental results using Learned World Models**, and propose a concrete **LLM experiment** designed to solve the computational overhead via In-Context Learning.

### 1. Clarification: The "Safe Seed Data" Assumption
Reviewers wHeb and KBsM questioned the realism of assuming access to a dataset free of reward hacking. We clarify that this assumption is necessary for **identifiability**: without a signal indicating the intended objective, it is impossible for *any* algorithm to distinguish between genuine task progress and reward hacking. We discuss multiple ways of collecting such datasets, including expert demonstrations, manual review and filtering, using simulators, and controlled lab environments.

In the context of Foundation Models, this "seed data" training corresponds to the **Supervised Fine-Tuning (SFT)** stage. Before an agent is optimized against a proxy reward (RLHF), it is supervised on expert demonstrations. MCVL acts as a safeguard during the subsequent RL phase, ensuring that the aggressive optimization of the proxy reward does not violate the priors learned during the SFT phase. Therefore, the requirement for safe seed data is not an additional burden; it utilizes the standard pre-training data already present in most large-scale RL applications. We will update the paper to clarify this point.

### 2. New Experimental Result: MCVL works with Learned Transition Models
Reviewers 2vux and KBsM noted that relying on a ground-truth simulator is restrictive. To demonstrate robustness, we conducted a new experiment on the *Box Moving* environment where we replaced the simulator with a **Learned World Model**.

The transition model was trained on transitions collected by running a random policy in the *Full* environment for just 20 episodes. MCVL successfully prevents reward hacking using only this learned model and achieves Oracle performance, despite the model's imperfections. We will update the paper with this new result.

### 3. Proposed Experiment: Scaling to Foundation Models via In-Context MCVL
We acknowledge the concern regarding computational cost and "toy domains" (Reviewer 2vux). To demonstrate that MCVL scales to complex tasks without prohibitive overhead, we propose an experiment using **In-Context MCVL** on a code generation task.

**Problem Setting: Poisoned MBPP**
We simulate a specification gaming scenario using the MBPP (Mostly Basic Python Problems) dataset, where 3 public tests are visible to the model. For a subset of problems, we modify the test suite to include incorrect tests (e.g., `sum(2, 2) == 5`). The proxy reward gives +1 for passing these incorrect tests, encouraging the model to hard-code incorrect logic (hacking). "True Performance" is measured on a hidden set of correct test cases.

**Method: In-Context Forecasting**
We replace the expensive gradient-based inner training loop with **In-Context Learning (ICL)**. This reduces the "Forecast" step to a single inference request:

1.  **Forecast (ICL):** We prompt the model with a batch of its own solutions and their advantages from the current iteration. We then ask it to generate a solution for a held-out validation problem while "adopting the style and strategies" of the high-advantage examples. This effectively simulates the behavior the model *would* converge to if it updated on those samples.
2.  **Score (Self-Reflection):** The agent acts as its own evaluator based on its SFT priors. It compares the **Forecasted Solution** (simulated update) against a **Zero-shot Solution** (current policy).
3.  **Gating:** The model is asked a binary question: *"Does the proposed solution represent a behavioral shift you wish to accept?"* If the forecasted behavior violates the model's internal concept of general correctness (e.g., by hard-coding answers to pass tests), the update batch is rejected.

---

> ### Author Response · Authors · 2025-11-27
>
> **Baselines:** We will compare In-Context MCVL against standard GRPO and GRPO with high KL-penalty.
> This design solves the *cost issue* (inference is cheaper than gradient steps) and the *complexity issue* (demonstrating efficacy on a code generation benchmark). It also highlights the core benefit of MCVL: a capable agent (LLM) in our setup is not incentivized to engage in reward hacking or deception. If it currently values "general helpfulness" (from SFT), it is a rational decision for it to reject an update that promotes hard-coding, even if that update yields high external reward.
>
> **We invite feedback on this design:** Do the reviewers consider the "Poisoned MBPP" task a sufficient proxy for measuring reward hacking in complex domains? Do we need to run any additional baselines?

---

### Note · Authors · 2025-12-04

I have read and agree with the venue's withdrawal policy on behalf of myself and my co-authors.